# The Additional 15 nt of 5′ UTR in a Novel Recombinant Isolate of Chilli Veinal Mottle Virus in *Solanum nigrum* L. Is Crucial for Infection

**DOI:** 10.3390/v15071428

**Published:** 2023-06-23

**Authors:** Qionglian Wan, Kaiyue Zheng, Jian Wu, Shan Bu, Mengting Jiao, Huijie Zhou, Yuwen Lu, Hongying Zheng, Guanwei Wu, Shaofei Rao, Hairu Chen, Fei Yan, Jiejun Peng

**Affiliations:** 1College of Plant Protection, Yunnan Agricultural University, Kunming 650201, China; wanqionglian@yxnu.edu.cn; 2State Key Laboratory for Managing Biotic and Chemical Threats to the Quality and Safety of Agroproducts, Institute of Plant Virology, Ningbo University, Ningbo 315211, China; kfz123456789@163.com (K.Z.); wujian@nbu.edu.cn (J.W.); bushan@hnu.edu.cn (S.B.); mengtingjiao96@163.com (M.J.); zhouhuijie1999@163.com (H.Z.); luyuwen@nbu.edu.cn (Y.L.); zhenghongying@nbu.edu.cn (H.Z.); wuguanwei@nbu.edu.cn (G.W.); raoshaofei@nbu.edu.cn (S.R.); 3Key Laboratory of Biotechnology in Plant Protection of MARA and Zhejiang Province, Institute of Plant Virology, Ningbo University, Ningbo 315211, China; 4School of Chemistry, Biology and Environment, Yuxi Normal University, Yuxi 653100, China

**Keywords:** chilli veinal mottle virus, potyvirus, recombination, genetic diversity, infectious clone

## Abstract

An isolate of chilli veinal mottle virus (ChiVMV; genus *Potyvirus*) of *Solanum nigrum* L. from southwest China (ChiVMV-YunN/Yuxi) was identified and sequenced (GenBank: OP404087). Comparison with other ChiVMV isolates and recombination analyses suggested a recombinant origin. The most significant recombination event among all 21 complete ChiVMV isolates was an ending breakpoint at 1408–1488 for ChiVMV-YunN/Yuxi with ChiVMV-TaiW and ChiVMV-YunN/Ca operating as the respective major and minor parents. Interestingly, the 5′ UTR of ChiVMV-YunN/Yuxi is 15 nucleotides (‘AAAAATAAAACAACC’) longer than other reported isolates. A full-length clone of ChiVMV-YunN/Yuxi was constructed and was shown to be infectious in *Nicotiana benthamiana*. The additional 15 nt of 5′ UTR in ChiVMV-YunN/Yuxi was stable when transmitted through three generations. Experiments with modified clones showed that the additional 15 nt are essential for infection by this isolate.

## 1. Introduction

Chilli veinal mottle virus (ChiVMV) is a member of the genus *Potyvirus* from the family Potyviridae and was first reported in West Malaysia in 1979 [1]. It has now become prevalent among several economically important crops in Asia including chilli, tomato and tobacco [2], causing symptoms of mosaic, distortion and vein banding on the leaves [1]. The ChiVMV genome has 9.7 kb of single-stranded RNA, excluding a poly (A) tail, which encodes a polyprotein that is cleaved into 10 mature functional proteins by virus-encoded proteases [3,4].

Genetic recombination is common in potyvirus populations [5]. Known ChiVMV isolates cluster with high probability into two clades [6] and recombination between isolates appears to be a significant factor affecting their diversity [6,7,8]. Coat protein (CP) gene sequences are used to estimate genetic diversity and to identify recombination events [6]. Recombination analysis revealed that the CP gene of isolate KX236451 is likely a recombinant between Pakistani isolate KT876050 (as the major parent) and the Indian isolate JN692501 (as the minor parent) [8]. Previous work in our laboratory predicted that two Chinese and one Pakistan isolate are recombinants and that recombination contributes to the variation in the CP sequences [7]. The CP to 3′ UTR region appears to be a hotspot for recombination among ChiVMV isolates [7,8]. Curiously, recombination has not been reported to be a major factor in the regions of the ChiVMV genome with the greatest overall diversity (P1, P3 and 6K2 coding regions).

Protein translation is the pivotal process in viral life cycles. Most cellular mRNAs rely on canonical mechanisms to initiate protein synthesis [9]. These mRNAs are modified at the 5′ end with a methyl guanylate cap and 3′ end with a poly(A) tail (a stretch of adenine nucleotides with a variable length) [10,11]. Potyviruses mainly rely on cis-acting elements in their 5′ UTR to initiate the translation of their polyprotein [12]. The 5′ UTRs of potyviruses can function as internal ribosomal entry sites (IRESes) without the need to scan from the 5′ end of their multi-AUG UTRs [13]. The 5’ UTR is responsible for conferring cap-independent translation, even onto reporter mRNAs. The stable stem–loop (SL) structures of the 5′ UTR of tobacco etch virus (TEV) serve as cap-independent regulatory elements (CIREs) to identify the elements responsible for its regulatory function and promote cap-independent translation [14]. Therefore, the 5′ UTRs of TEV and pea seedborne mosaic virus (PSbMV) RNAs are found to significantly increase gene expression [15]. Mutational analysis of the 5’ UTR of potato virus Y (PVY) is conducted to find the last 55 nucleotides necessary for the enhancement, and the 5’ UTR may play a very important role in the translational enhancement in plant cells [16].

In this study, we report a new recombinant ChiVMV-YunN/Yuxi (GenBank: OP404087) from *Solanum nigrum* L., which has a recombinant event in its P1 region. In addition, the 5′ UTR of ChiVMV-YunN/Yuxi is 15 nucleotides longer than the other reported isolates and we show that these additional nucleotides are important for viral infection.

## 2. Materials and Methods

### 2.1. Sample Collection, De Novo Assembly and RACE

During a local virus survey in 2021, 40 samples of *S. nigrum* plants with virus-like symptoms were collected from Huang Caoba village in Hongta district (Figure 1). To assess the viral species present in the viral samples, 10 samples of leaves were mixed into a pool (4 pools were obtained) and the total RNA was extracted from mixed fresh leaves using a TRIzol reagent (Invitrogen). The mRNA was purified from total RNA using poly-T oligo-attached magnetic beads. RNA integrity was checked using an Agilent 2100 Bioanalyzer (Agilent Technologies). The TruSeq RNA Sample Preparation Kit (Illumina) was used to construct cDNA libraries according to the manufacturer’s instructions. The high-throughput transcriptome sequencing was performed on an Illumina NovaSeq 6000 platform with PE150 bp and a CLC Genomics Workbench 20 (QIAGEN) was used for sequencing and data analysis. The contigs were generated de novo and compared with amino acid sequences in GenBank using BLASTx.

To obtain the full-length sequence of ChiVMV, the 3′-end first strand cDNA was synthesized with M4T primers using ReverTra Ace™ qPCR RT Kit (Toyobo) according to the manufacturer’s instructions (Appendix A and Figure 2A,B), and the PCR products were then amplified with M4 and 3’-RACE-F using KOD-plus-Neo (Toyobo) [17,18]. For 5′ RACE, the first strand cDNA was synthesized with a gene-specific reverse primer 5’-RACE-ChiVMV-YunN-R and the adapter primer ZHM1 was ligated to cDNA/RNA duplexes using T4 RNA ligase (Takara) (Appendix A). The PCR product of 5′ RACE was amplified with ZHM2 and 5’-RACE-R, and then cloned and sequenced [19]. To ensure the integrity and authenticity of full-length sequences, two overlapping sequence fragments were amplified with the primer pairs ChiVMV-2L-1 and ChiVMV-2L-2 (Appendix A).

### 2.2. Viral Recombination Analysis

The recombination detection program RDP5 was used to predict recombination events among all 21 full-length sequences (20 in NCBI and 1 in this study) (Appendix A). The program simultaneously uses a range of different recombination detection methods to detect recombination events within aligned sequences. These methods include the BOOTSCANNING method, the GENECONV method, the maximum chi-square method (MAXCHI), the CHIMAERA method, the sister-scanning method (SISCAN), the 3SEQ method, and the VisRD method [20]. Recombinant events that were identified by at least three methods and *p* values of less than 1 × 10^−6^ were considered credible.

### 2.3. Phylogenetic and Sequence Analysis

Complete genome sequences of ChiVMV were obtained from GenBank and this study (Appendix A). Pairwise nucleotide sequence comparisons were performed using the SDT (Species Demarcation Tool) v1.2 program. The evolutionary history was inferred using the maximum likelihood method [21]. The best-fit nucleotide substitution model was determined to be neighbor-joining by MEGA X [22]. Evolutionary analyses were conducted in MEGA X with 1000 bootstrap replicates.

### 2.4. Plasmid Construction and Agroinfiltration

RT-PCR was performed using KOD-plus-Neo (Toyobo) and in accordance with the manufacturer’s protocol. PCR products were purified with the Gel Extraction Kit (Omega, Norcross, GA, USA). To generate infectious clones, the ClonExpress II One Step Cloning Kit (Vazyme) was used for homologous recombination. The full-length ChiVMV was amplified with a primer pair and recombined with the linearized binary vector pCB301-MD [1]. To confirm infectivity, the infectious clones were transformed into *Agrobacterium tumefaciens* and inoculated as described before [1]. The mGFP tagging and mutagenesis of clones were also constructed via homologous recombination and the primer combinations were listed in Appendix A (Appendix A).

### 2.5. Virion’s Purification

Leaves were homogenized in an extraction buffer (0.05 M K_2_HPO_4_; 0.01 M Na_2_-EDTA; 1% (*w*/*v*) Na_2_SO_3_; 5% (*v*/*v*) ethanol adjusted to pH 7.6) using 3 mL per gram of leaves. The homogenate was filtered through preboiled cheesecloth, centrifuged at 8000× *g* for 30 min, and the pellet discarded. Triton X-100 was added to the supernatant to a final concentration of 1% (*v*/*v*). This was stirred gently for 1 h and then centrifuged for 3 h at 90,000× *g* on a sucrose cushion of 20% (*w*/*v*) sucrose in an extraction buffer, filling approx. 25% of each tube. The supernatant was discarded, and the pellet was suspended in the buffer (the same as the extraction buffer but without ethanol) via overnight soaking.

### 2.6. Sap Inoculation and Western Blot

We cut symptomatic young leaves from the infected plants and ground them in a 1 mL phosphate buffer. We dipped segments of our forefingers into the sap and rubbed them gently over the leaves. Immediately after inoculation, we rinsed the leaves with double-distilled water. Then, we transferred the plants to the glasshouse.

Reverse transcription (RT) was performed using the ReverTra AceTM qPCR RT Master Mix (Toyobo) and a polymerase chain reaction (PCR) was performed by KOD-plus-Neo (Toyobo). RT was performed at 42 °C for 60 min with M4T followed by 72 °C for 10 min. The cycling conditions for the subsequent PCR were: 98 °C 5 min, then 32 cycles of 98 °C for 30 s, 55 °C for 30 s, 68 °C for 1 kb/min; and 68 °C for 10 min. The additional 15 nt was detected with primer pairs UTR f/r (with 15 nt) or UTR_d15_ f/UTR r (without 15 nt) (Appendix A).

The total proteins of *N. benthamiana* leaves (100 mg) were extracted with a protein extraction buffer (50 mmol L^−1^ sodium phosphate buffer pH 7.0, 5 mmol L^−1^ β-mercaptoethanol, 10 mM EDTA, 0.1% Triton X-100), mixed with 5× loading buffer and separated by 12% SDS polyacrylamide gel electrophoresis (PAGE). Proteins were transferred onto nitrocellulose membranes (Amersham) via transfer buffer (48 m mmol L^−1^, Tris-base, 39 mmol L^−1^, glycine, 20% methyl alcohol). Rabbit anti-ChiVMV-CP (AtaGenix, Wuhan, China) or mouse anti-GFP primary antibodies (Sigma-Aldrich, St. Louis, MO, USA) diluted 1:5000 was used for Western blotting, followed by horseradish peroxidase-coupled goat anti-rabbit/mouse IgG (Sigma-Aldrich) diluted 1:5000.

## 3. Results

### 3.1. Sequence, Phylogenetic and Recombinant Analysis of ChiVMV-YunN/Yuxi from Solanum nigrum L. in Yuxi City

Recently, ChiVMV has often been detected in solanaceous crops in Yunnan province, China. Virus-like symptoms (interveinal leaf crinkle, vein distorting and systemic mosaic) were also observed in *Solanum nigrum* L. This is a weed found in and around chilli fields and which is also grown as a Cd hyperaccumulator plant for remediating Cd-polluted soil [23] throughout Yuxi city, Yunnan province (Figure 1A). The typical symptoms of 40 samples were collected from two sites in Huang Caoba village in Hongta district. To assess the viral species present in viral samples, 10 samples of leaves were mixed into a pool (4 pools were obtained) and sent for next-generation RNA-sequencing (NGS). The paired-end clean reads of four pools (a: 21,493,340, b: 21,789,198, c: 21,190,015 and d: 22,420,959) were obtained, and the contigs of four pools (a: 35,065, b: 37,595, c: 36,027 and d: 40,599) were generated de novo and compared with sequences in the GenBank aa using BLASTx. In total, 25 contigs (a; b; c and d) were identified with E-values of zero. The contigs of *S. nigrum* L. were matched with tomato yellow mottle-associated virus (TYMaV, 16 contigs), tobacco vein banding mosaic virus (TVBMV, 2 contigs), chilli veinal mottle virus (ChiVMV, 5 contigs) and cabbage cytorhabdovirus 1 (CCyV-1, 1 contig), respectively.

In order to confirm the suspected presence of ChiVMV, we used primer pair Det-ChiVMV f and Det-ChiVMV r to amplify a partial fragment of the CP by RT-PCR (Appendix A). Fragments with a predicted size of approximately 581 bp were obtained in 28 of the 40 samples. To confirm the infectivity of the virus, *Nicotiana benthamiana* plants were mechanically inoculated with sap from *S. nigrum* leaves (Figure 1B). Western blot analysis confirmed the presence of ChiVMV in the systemic (non-inoculated) leaves of inoculated plants (Figure 1C). Virions were purified and visualized via negative stain transmission electron microscopy (TEM) (Figure 1D). Then, viral RNA was confirmed by RT-PCR and a 1153 bp sequence was obtained which had a 93.8% nt identity to ChiVMV-GD (GenBank: KU987835.1). To ensure the integrity and authenticity of the full-length sequence of ChiVMV, two overlapping sequence fragments were amplified with the primer pairs ChiVMV-2L-1 and ChiVMV-2L-2 (Appendix A). The RT-PCR products were expected to be sizes 6859 bp and 6674 bp, thus overlapping by 3792 bp (Appendix A). The complete sequence was 9741 nt long (GenBank: OP404087), and we tentatively designated this virus as chilli veinal mottle virus isolate Yunnan Yuxi (ChiVMV-YunN/Yuxi).

The complete genomic RNA sequence of ChiVMV-YunN/Yuxi has a 5′ UTR (nts 1–186) and 3′ UTR (nts 9457–9741), with a single large open reading frame (ORF) encoding a polyprotein of 3089 aa residues. To investigate the phylogenetic relationships between ChiVMV-YunN/Yuxi and other ChiVMV (es), the full genome sequences of 20 ChiVMV isolates were retrieved from GenBank, and a NJ phylogenetic tree was inferred by Mega X. The isolates clustered into two groups, with ChiVMV-YunN/Yuxi in Group I (Figure 2A). ChiVMV-YunN/Yuxi had an 84.0–89.7% nucleotide identity to other Group I sequences and an 82.2–86.1% identity to isolates of Group II (Figure 2B). Among the different genome regions, the greatest differences between isolates were found in P1 (Figure 2B). The 5′ part (5′ UTR to P1) of ChiVMV-YunN/Yuxi had the highest nucleotide identity (95.7%) to the corresponding region of ChiVMV-YunN/Ca (Group II), but the 3′ part (HC-Pro to 3′ UTR) was most closely related (92.5% nucleotide identity) to ChiVMV-TaiW in Group I (Appendix A). A phylogenetic analysis also showed that the 5′ part of ChiVMV-YunN/Yuxi clustered into Group II, but that the 3′ part clustered in Group I (Appendix A). This suggested that ChiVMV-YunN/Yuxi might be a recombinant between Group I and Group II isolates.

Recombinant events in ChiVMV have been reported previously [7] and we therefore used RDP5 software to predict recombination events among all 21 full-length sequences. There were 18 credible recombinant events in total and the top 10 events are listed in Appendix A. Of these, the most significant was an ending breakpoint at 1408–1488 for ChiVMV-YunN/Yuxi, with ChiVMV-TaiW and ChiVMV-YunN/Ca serving as the respective major and minor parents (Figure 2C). This was consistently identified using GENECONV (*p* value of 1.743 × 10^−109^), RDP (3.518 × 10^−113^), BootScan (2.247 × 10^−107^), MaxChi (6.616 × 10^−35^), Chimera (3.289 × 10^−36^), SiScan (2.440 × 10^−48^) and 3Seq (5.668 × 10^−111^) (Appendix A). These results suggest that ChiVMV-YunN/Yuxi is a novel recombinant ChiVMV isolate.

### 3.2. ChiVMV-YunN/Yuxi Has a Longer 5′ UTR Than Other Isolates That Is Stable in Subculture

Interestingly, the 5′ UTR of ChiVMV-YunN/Yuxi is 15 nucleotides (‘AAAAATAAAACAACC’) longer than other reported isolates (Appendix A). To confirm the presence of additional 15 nt in 5′ UTR, virus infected-plants were collected, and RT-PCR confirmed the presence of 5′ UTR, with an additional 15 nt in the systemic (non-inoculated) leaves of inoculated plants (Figure 1E). Concurrently, an infectious clone of ChiVMV-YunN/Yuxi was constructed for further investigation. The full-length cDNA infectious clone (pChiVMV) was transformed into *A. tumefaciens*, which was then delivered to *N. benthamiana* plantlets by infiltration. Severe mosaic was detected in systemic leaves at 5 days post-infiltration (Figure 3A). The flexuous filamentous virions were observed in new non-inoculated leaves by TEM (Figure 3B) and Western blot confirmed that the virus had spread systemically in the inoculated plants (Figure 3C). These results demonstrate the infectivity of the full-length ChiVMV-YunN/Yuxi. The virus was then transmitted by sap inoculation for three generations, and all plants developed severe mosaic symptoms in their systemic leaves 4 days after inoculation (Figure 3D). RT-PCR also confirmed that the additional 15 nt in the 5′ UTR were stable during this subculture (Figure 3E).

### 3.3. The Additional 15 nt in the 5′ UTR Are Crucial in Viral Infection

To trace and observe infection by ChiVMV-YunN/Yuxi more conveniently, an mGFP-tagged cDNA infectious clone was then constructed. The sequence encoding a NIa protease cleavage site (EGGEVTHQ/SG) was introduced between the eGFP and CP coding sequences (Appendix A). In order to study the biological functions of 15 nt in ChiVMV-YunN/Yuxi 5′ UTR, the vector of pChiVMV_d15_-GFP was constructed. This was identical to pChiVMV-GFP, but lacked the 15 nt sequence. The infectious clones pChiVMV-GFP and pChiVMV_d15_-GFP were transformed into *A. tumefaciens* and then delivered to *N. benthamiana* plantlets by infiltration. At 5 dpi, UV lamp examination showed GFP fluorescence in the inoculated leaves of pChiVMV-GFP infected plants. At 7 dpi, GFP fluorescence was observed in the upper non-inoculated leaves of pChiVMV-GFP infected plants, but not in those infected by pChiVMV_d15_-GFP (Figure 4A). Western blots also showed that the coat protein and GFP could be detected in the new non-inoculated leaves, but only those of pChiVMV-GFP-infected plants (Figure 4B).

To further confirm the function of the additional 15 nt and to investigate the significance of the A residues, another three mutant infectious clones were constructed (Appendix A). In pChiVMV_MutC_-GFP, the A bases were all replaced with C, while the other clones underwent the addition or deletion of an A residue at the N-terminus (pChiVMV_Add1A_-GFP and pChiVMV_delete1A_-GFP, respectively). pChiVMV-GFP and the three mutant infectious clones were delivered to *N. benthamiana* plantlets by A. tumefaciens. At 7 dpi, GFP fluorescence was observed in the upper non-inoculated leaves of plants infected with pChiVMV-GFP and pChiVMV_Add1A_-GFP (Figure 5A). At 8 dpi, GFP fluorescence was observed in 3/3 upper non-inoculated leaves of plants infected with pChiVMV-GFP or pChiVMV_Add1A_-GFP, in 1/3 leaves of those infected with pChiVMV_MutC_-GFP, but not in those inoculated with pChiVMV_delete1A_-GFP (Figure 5A,B). Western blots confirmed that there were significant differences in the accumulation levels of virus proteins in the order: pChiVMV-GFP > pChiVMV_Add1A_-GFP > pChiVMV_MutC_-GFP > pChiVMV_delete1A_-GFP (Figure 5). By 12 dpi, the GFP fluorescence was observed in all the infected plants. The results show that these 15 nt of the 5′ UTR are crucial for infection by ChiVMV-YunN/Yuxi.

## 4. Discussion

Molecular criteria are often used in the taxonomy of potyviruses. Isolates of different species typically have sequence identities of their large ORF (or, if necessary, the CP-coding region) < 76% (nt) and <82% (aa) [24]. Thresholds for other coding regions, ranging from 58% (P1 coding region) to 74–78% (in other regions), have become commonly accepted [25]. In this study, we have identified a novel recombinant ChiVMV isolate taken from *S. nigrum*. There are significant sequence differences among the 21 ChiVMV isolates, but the variation remains within the limits expected for isolates of the same species. The genetic variations were not distributed evenly among the different regions of the ChiVMV genome, with the greatest diversity seen in the P1, P3 and 6K2 coding regions. Recombination is important in the evolution, sequence diversity and complex genotyping system of potyviruses [26]. Earlier investigations demonstrated that the CP to 3′ UTR region was a hotspot for recombination in ChiVMV isolates [7,8]. In this study, recombination events were identified in most of the genomic regions of ChiVMV, with a higher frequency in the P1 to HC-Pro and 6K2 to VPg regions (Appendix A) and a significant recombination hotspot in the P1 region (3/10 events). P1 may be under lesser evolutionary constraints than the other proteins, making it a mutational hotspot in ChiVMV.

Previous studies show that the 5′ UTRs of potyviruses can significantly increase gene expression in plant cells. The SL structures and last 55 nt of the 5′ UTR are necessary for this enhancement [16]. In this study, the first 15 nt of ChiVMV-YunN/Yuxi RNA promoted efficient viral infection (Figure 4 and Figure 5). However, there were no predicted significant secondary RNA structures in the first 48 nt (Appendix A), suggesting that secondary structure is not a factor in this effect. Previous studies showed that the first 16 bases of PVY (potato virus Y) RNA have an essential function in the initiation of translation. Antisense oligonucleotide A (16-1) mediated the inhibition of translation, even when the concentration of A (16-1) was reduced to 25/1 of the transcript 5′ PVY-GUS [1]. The first 15 nt of ChiVMV-YunN/Yuxi RNA appear to play a crucial role in viral infection, but the mechanism of enhancement is unclear and may play a similar role to the PVY (1–16 nt).

In the protein translation of potyviruses in general, the viral VPg, rather than the canonical 5′ cap, interacts with eukaryotic translation initiation factor 4E (eIF4E) or its isoform iso 4E [eIF(iso)4E] [27]. Previous studies have shown that the ChiVMV VPg interacted with both eIF4E and eIF(iso)4E and that silencing of *eIF4E* and *eIF(iso)4E* decreased the accumulation of ChiVMV [28]. The translation of potyviruses may be stimulated by a collaboration between the 5′ UTR and VPg. The cap structure at the 5′ ends of mRNAs has been shown to play an important role in the initial ribosome entry and may also protect RNAs against exonuclease degradation. When the 5′ PVY-GUS was capped, the translation efficiency was sixfold higher than in conditions with uncapped mRNA [29]. We next plan to assess whether the first 15 bases to help link to the VPg or canonical cap structure to promote translation.

## Figures and Tables

**Figure 1 viruses-15-01428-f001:**
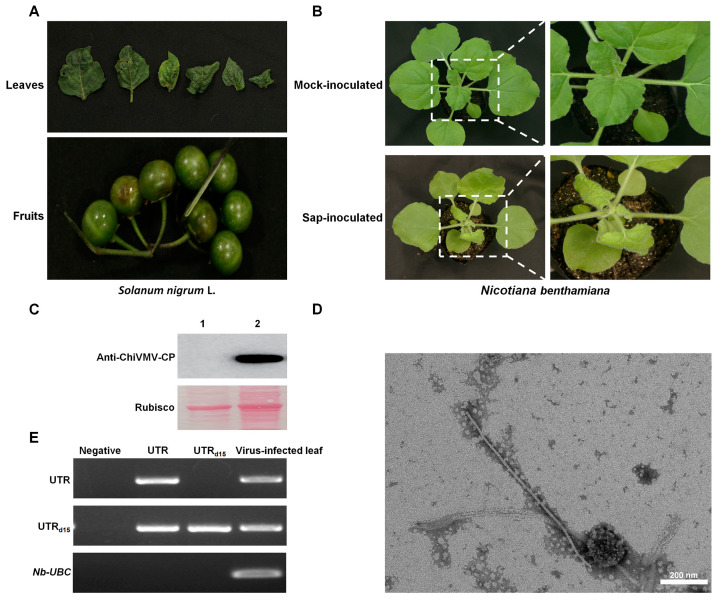
Symptoms of virus-infected *Solanum nigrum L.* and *Nicotiana benthamiana*. (**A**) Symptoms of virus-infected *S. nigrum L.* from field; (**B**) symptoms on plants of *N. benthamiana* inoculated with water or viral sap at 10 dpi; (**C**) Western blot confirming the presence of ChiVMV in systemic leaves of inoculated plants. 1. Systemic leaf of Mock-inoculated plant; 2. systemic leaf of sap-inoculated plant; (**D**) virions observed by TEM in negatively stained samples of systemic leaves of *N. benthamiana* plants. Bars represent 200 nm; (**E**) RT-PCR confirming the presence of the additional 15 nt of 5′ UTR in systemic leaves of inoculated plants (UTR: with 15 nt; UTR_d15_: without 15 nt).

**Figure 2 viruses-15-01428-f002:**
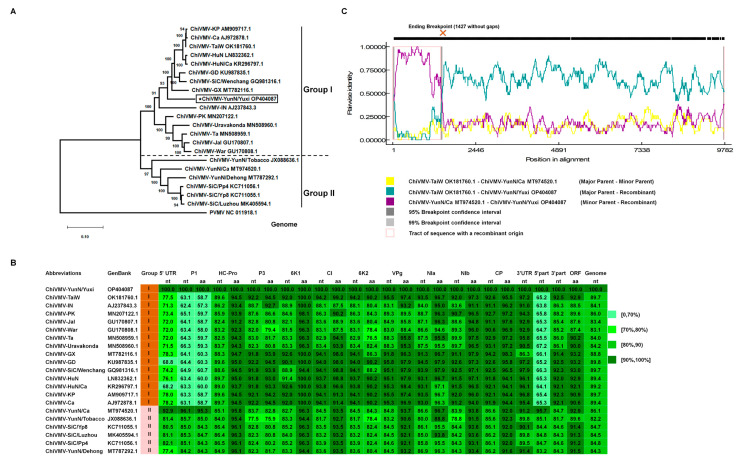
Phylogenetic analysis of ChiVMV-YunN/Yuxi. (**A**) Neighbor-joining phylogenetic trees constructed using MEGA X showing the relationship of ChiVMV-YunN/Yuxi to other full-length genomes of ChiVMV (es) using nucleotide sequences. Pepper veinal mottle virus (PVMV, GenBank: NC011918.1) was used as an outgroup. Numbers on branches are bootstrap support values (1000 replicates). We employed the maximum composite likelihood + G model. The virus abbreviations and accession numbers are listed in Appendix A. ★: Sequence determined in this study. (**B**) Amino acid (aa) and nucleotide (nt) sequence the identities of the 10 mature proteins and nucleotide identities of the untranslated regions (UTR) of ChiVMV-YunN/Yuxi with those of other isolates. Increasing levels of saturation of green denote high and low levels of amino acid similarity. (**C**) Recombination (RDP) and genetic map of ChiVMV-YunN/Yuxi showing major (ChiVMV-TaiW) and minor (ChiVMV-YunN/Ca) parents and the recombinant region was highlighted in pink. Multiple nucleotide sequences were aligned using MUSCLE and then analyzed using RDP5 with default parameters.

**Figure 3 viruses-15-01428-f003:**
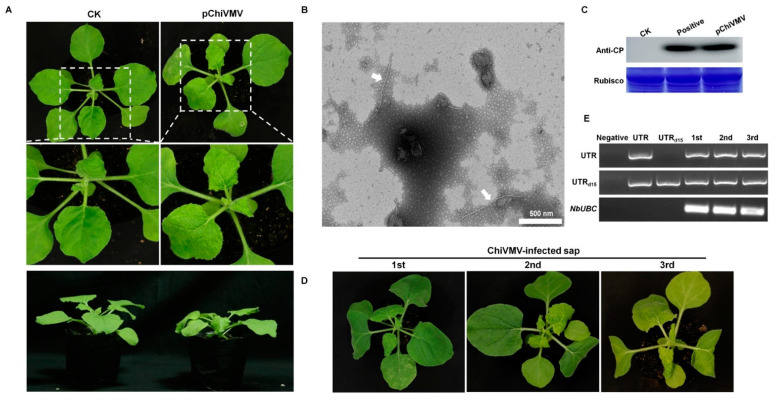
Infectivity and symptoms of ChiVMV-YunN/Yuxi following inoculation of *Nicotiana benthamiana* by pChiVMV-YunN/Yuxi. (**A**) Phenotype of *N. benthamiana* plants agroinfiltrated with viral infectious clone combinations or empty agrobacterium (CK) at 5 days post-infiltration. (**B**) Typical potyvirus virions in negatively stained samples of systemic leaves of plants inoculated with the ChiVMV-YunN/Yuxi infectious clone. Bars represent 500 nm. White arrows indicate the virions. (**C**) Western blot confirming the presence of ChiVMV in systemic leaves of inoculated plants. (**D**) Phenotype of *N. benthamiana* plants inoculated with ChiVMV-YunN/Yuxi infected saps after three consecutive subcultures. (**E**) RT-PCR confirming the presence of 15 nt in systemic leaves of inoculated plants.

**Figure 4 viruses-15-01428-f004:**
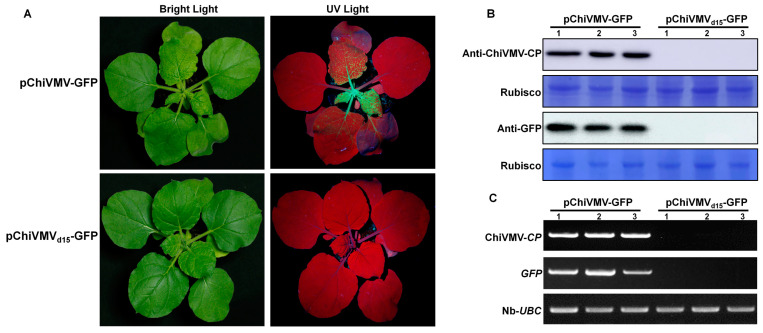
Expression of eGFP by the infectious clone pChiVMV-GFP in *N. benthamiana*. (**A**) Symptoms in *N. benthamiana* plants inoculated with pChiVMV-GFP or pChiVMV_d15_-GFP (where first 15 nt of pChiVMV-GFP genome were deleted) under UV and natural light at 7 dpi. (**B**) Western blot analysis confirming the presence of ChiVMV-GFP or ChiVMV_d15_-GFP in the infected leaves of *N. benthamiana* plants (3 plants/treatment). (**C**) RT-PCR analysis confirming the presence of ChiVMV-GFP or ChiVMV_d15_-GFP in the infected leaves of *N. benthamiana* plants.

**Figure 5 viruses-15-01428-f005:**
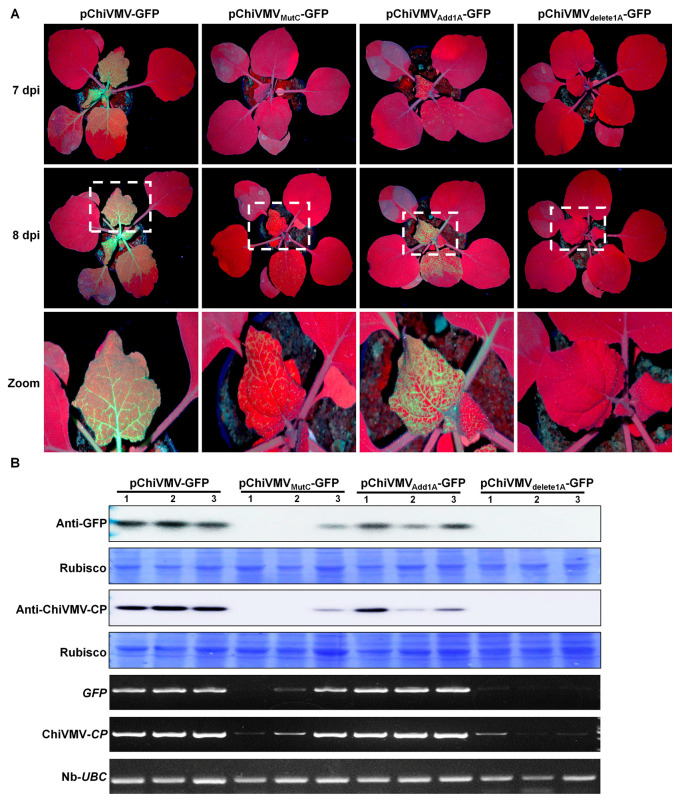
Expression of eGFP by the mutant infectious clone pChiVMV-GFP in *N. benthamiana*. (**A**) Symptoms in *N. benthamiana* plants inoculated with pChiVMV-GFP, pChiVMV_MutC_-GFP (where the A base of first 15 nt of pChiVMV-GFP were replaced with C), pChiVMV_Add1A_-GFP (where one A base was added to the 5′ end of pChiVMV-GFP genome) or pChiVMV_delete1A_-GFP (where the first A base of pChiVMV-GFP was deleted) under UV and natural light at 7–8 dpi. (3 plants/treatment) (**B**) Western blot and RT-PCR analysis confirming the presence of viruses’ coat protein in the infected leaves of *N. benthamiana* plants.

## Data Availability

The data presented in this article are available on request from the corresponding authors.

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
