# Peer review of "The Additional 15 nt of 5′ UTR in a Novel Recombinant Isolate of Chilli Veinal Mottle Virus in Solanum nigrum L. Is Crucial for Infection"

_viruses, 2023, doi:10.3390/v15071428_

Round 1
Reviewer 1 Report
This manuscript, by Wan et al, reported a novel recombinant Chilli Veinal Mottle Virus (ChiVMV) isolate. The recombination analysis suggested that the most significant recombination event occurs at the P1 coding region, with major and minor parents from two separate genotype groups. Sequence alignment suggested that this novel recombinant isolate possessed a stable 15-nt insertion at the 5’-end of the genome, which was later revealed to be critical for viral infectivity. The results of this manuscript were clearly presented, however more details should be given in the section of materials and methods.
Major comments:
1. Line 51-60: I recommend pointing out the exact potyvirus species for each reference. This could be very helpful for readers.
2. Section 2.1: Figure S1 should be cited at least once in this section as this supplementary figure can clearly demonstrate the 5’RACE and 3’RACE assays described in line 77-81.
3. More details should be given in the materials and methods section:
a. How was the full-length genome sequenced
b. How was the cloning performed, including mGFP tagging and mutagenesis
c. How was the sap inoculation performed
d. How was the western blot performed, and what antibodies were used for detection
e. How was the detection of the additional 15 nt by RT-PCR performed
4. Line 283-286, “a similar role to the PVY…”: This sounds like the role of the 15-nt addition of the novel ChiVMV isolate in protein translation is tested in this study. Please consider rephrasing this sentence.
Minor comments:
1. Line 77, “3’-RACE-R”: do you mean 3’-RACE-F”?
2. Line 139, “strain”: This is a typo and should be “stain”.
3. Figure S3 is not cited anywhere in the main text. I think it should be cited around line 194?
Some minor English grammar errors were found. For example, line 109 and line 119.
Author Response
Author’s response to comments received
We thank the editor and the reviewers for their constructive comments and have now added the request experiments and modified the manuscript to correct the errors and to respond to the comments and suggestions. In the text below, the comments received are in black and our replies in red.
Reviewer 1:
Major comments:
1. Line 51-60: I recommend pointing out the exact potyvirus species for each reference. This could be very helpful for readers.
Reply: We thank the reviewer for the useful comments. The information has been added at Lines 56-60.
2. Section 2.1: Figure S1 should be cited at least once in this section as this supplementary figure can clearly demonstrate the 5’RACE and 3’RACE assays described in line 77-81.
Reply: Cited (Lines 85 and 88).
3.More details should be given in the materials and methods section:
a.How was the full-length genome sequenced?
Reply: The information was added (Lines 88-90).
b.How was the cloning performed, including mGFP tagging and mutagenesis?
Reply: The information was added (Lines 114-116).
c.How was the sap inoculation performed?
Reply: The information was added (Lines 127-130).
d.How was the western blot performed, and what antibodies were used for detection?
Reply: The information was added (Lines 138-146).
e.How was the detection of the additional 15 nt by RT-PCR performed?
Reply: The information was added (Lines 131-137).
4. Line 283-286, “a similar role to the PVY…”: This sounds like the role of the 15-nt addition of the novel ChiVMV isolate in protein translation is tested in this study. Please consider rephrasing this sentence.
Reply: We have now rewritten this sentence (Lines 320-322).
Minor comments:
1. Line 77, “3’-RACE-R”: do you mean 3’-RACE-F”?
Reply: Corrected.
2. Line 139, “strain”: This is a typo and should be “stain”.
Reply: Corrected (Line 171).
3. Figure S3 is not cited anywhere in the main text. I think it should be cited around line 194?
Reply: Corrected (Line 228).
Comments on the Quality of English Language
Some minor English grammar errors were found. For example, line 109 and line 119.
Reply: Corrected.

Reviewer 2 Report
Authors Qionglian Wan and co-workers presented a manuscript entitled "Identification of a novel recombinant chilli veinal mottle virus in Solanum nigrum L. from Yuxi city, Yunnan province in China".
They identified and sequenced an isolate (designated ChiVMV-YunN/Yuxi) of chilli veinal mottle virus - ChiVMV (genus Potyvirus) from symptomatic Solanum nigrum plants in southwest China. Phylogenetic analysis suggested a recombination event, which was confirmed by the RDP5 programme. The recombination breakpoint was identified in a P1 region.
Sequencing also revealed a longer 5'UTR region. To analyse the properties of this region, the authors prepared a set of infectious clones mutated in the 5'UTR genomic region.
Suggestions for improving the manuscript:
The title of the manuscript is misleading. A better expression of the content of the manuscript can be achieved by adding the word "isolate" - Identification of a novel recombinant isolate of chilli veinal mottle virus in Solanum nigrum L. from Yuxi city, Yunnan province in China.
Another way to improve the title is to emphasise the results of the analysis of the 5'UTR region, which provides the most novel information.
Chapter 2.1. "Sample collection, de novo assembly and RACE" needs a more detailed description of the HTS procedure.
Page 3, lines 119-123: missing words, sentences are difficult to understand.
Page 3, line 124: information on pooling of samples should also be mentioned in a "Materials and methods" chapter.
Figures need better organisation of their subdivided parts and also more detailed captions to improve their understanding.
See previous comments.
Author Response
Author’s response to comments received
We thank the editor and the reviewers for their constructive comments and have now added the request experiments and modified the manuscript to correct the errors and to respond to the comments and suggestions. In the text below, the comments received are in black and our replies in red.
Reviewer 2:
1.The title of the manuscript is misleading. A better expression of the content of the manuscript can be achieved by adding the word "isolate" - Identification of a novel recombinant isolate of chilli veinal mottle virus in Solanum nigrum L. from Yuxi city, Yunnan province in China. Another way to improve the title is to emphasize the results of the analysis of the 5'UTR region, which provides the most novel information.
Reply: We thank the reviewer for the useful comments. We have revised the title.
2.Chapter 2.1. "Sample collection, de novo assembly and RACE" needs a more detailed description of the HTS procedure.
Reply: We added the description at Lines 73-76.
3.Page 3, lines 119-123: missing words, sentences are difficult to understand.
Reply: Corrected (Line 203).
4.Page 3, line 124: information on pooling of samples should also be mentioned in a "Materials and methods" chapter.
Reply: Added at Lines 70-72.
5.Figures need better organization of their subdivided parts and also more detailed captions to improve their understanding.
Reply: Revised in Figures and Figure legends.
